# Generating the Generator: A User-Driven and Template-Based Approach towards Analog Layout Automation

**Benjamin Prautsch \***, **Uwe Eichler and Uwe Hatnik**

Fraunhofer IIS/EAS, Institute for Integrated Circuits, Division Engineering of Adaptive Systems,
01187 Dresden, Germany

**\*** Correspondence: benjamin.prautsch@eas.iis.fraunhofer.de; Tel.: +49-351-45691-280

**Abstract:** Various analog design automation attempts have addressed the shortcomings of the still largely manual and, thus, inefficient and risky analog design approach. These methods can roughly be divided into synthesis and procedural generation. An important key aspect has, however, rarely been considered: usability. While synthesis requires sophisticated constraints, procedural generators require expert programmers. Both prevent users from adopting the respective method. Thus, we propose a new approach to automatically create procedural generators in a user-driven way. First, analog generators, which also create symbols and layouts, are utilized during schematic entry to encapsulate common analog building blocks. Second, automatic code creation builds a hierarchical generator for all views with the schematic as input. Third, the approach links the building block generators with the layout through an object-oriented template library that is accessible through generator parameters, allowing the user to control the arrangement. No programming is required to reach this state. We believe that our approach will significantly ease the transition of analog designers to procedural generation. At the same time, the templates allow for a "bridge" to open frameworks and synthesis approaches so that the methodologies can be both better spread and combined. This way, comprehensive frameworks of both synthesis-based and procedural-based analog automation methods can be built in a user-driven way, and designers are enabled to gain early layout insight and ease IP reusability.

**Keywords:** IC design; analog layout; reuse; EDA; design automation; generators; templates; usability; code generation





## 1. Introduction

Analog IC design still relies on largely manual design entry and manual design iterations. Despite a variety of automation attempts that have been demonstrated, only a few have found their way into the broadly accepted industrial design environments. Simulation and schematic-level optimization is mainstream. Procedural generators automate device-level layouts and layout tools support designers interactively during manual design entry, e.g., by schematic-driven design or on-line design rule checking. All these tools, however, do not automate actual design or help with reusing circuitry but rather accelerate individual design steps. The lack of comprehensive automation, such as in the digital domain, still sets analog design productivity far behind.

In order to overcome this shortcoming, we propose a new method that enables analog design engineers to create procedural generators on their own and, thus, ease IP reuse. The method is based on procedural automation that utilizes generators at hierarchy levels far above device level. In addition, we overcome the need for programming generator code by means of automatic code generation. This enables designers to get a flexible generator in a matter of minutes. This way, parametric layout automation is made available without the former need to wait for expert generator programmers. We believe that this approach will significantly lower the entry barrier for utilizing analog generators by diminishing the

former hurdle of initial generator programming. As a result, analog design engineers are enabled to leverage themselves and their individual analog designs.

### 1.1. Context of the Presented Work

Analog IC design can be broken down into many separate design steps that, however, are closely linked with each other. Much research has been conducted across these levels that would exceed the scope of this paper. Thus, we first clarify this work's scope.

Figure 1 shows the simplified analog design flow and depicts this work's focus on both schematic-level design entry and layout design. The automation approach is generator-based and, thus, is to be distinguished from synthesis methods (see below). While synthesis requires a sophisticated and complete set of constraints, generators must initially be programmed by EDA experts. Both constraints management and dedicated generator programming are significant entry barriers to the respective automation method. As an example, complex constraint management becomes adherent to enable synthesis methods [1–3], while generator development requires significant development time. The latter was even considered a "paradigm shift" by the Berkeley Analog Generator (BAG) working group [4].

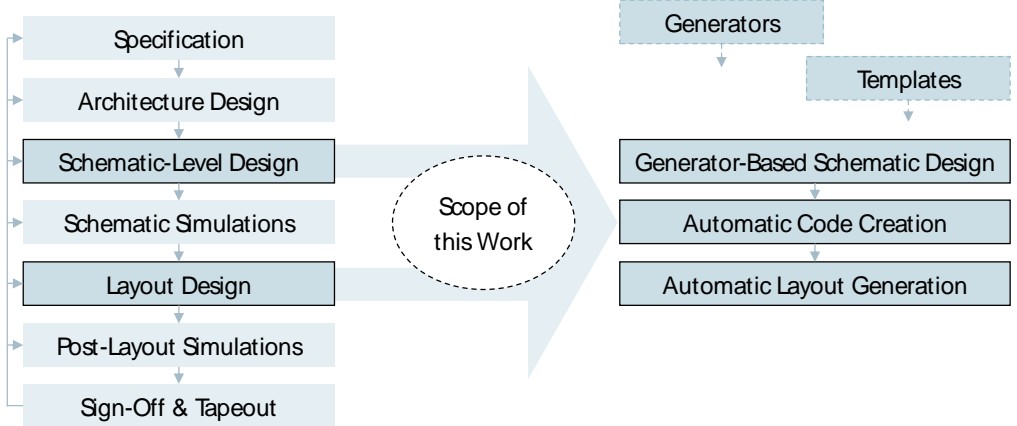

**Figure 1.** Scope of this Work. From the usual design flow, this work focusses on both schematic and layout design as well as the connection of them in a systematic and reusable way.

We believe that a combined approach of both synthesis methods and procedural automation will pave the way towards full analog automation—in accordance with [5]. As a step towards this goal, we address automatic code generation of procedural generators that automate layouts (plus schematic and symbol views) in a user-driven way, based on existing designs. In addition, we support various PDKs through abstraction of technology data [6].

### 1.2. Our Contribution

We pursue increased flexibility of otherwise structurally relatively static analog layout generators. Former generators can only be adapted to new requirements by time-consuming programming. This work combines several recent advances in generator-based layout automation and implements a user-driven method that allows automatic generator creation in a matter of seconds. The new method utilizes the following recent advances:

- Generator-based schematic design entry and cellview generation of building blocks that include the views symbol, schematic, and layout [7],
- Template-based extensions of generators in a matrix style [8],
- Template-based extensions of generators in a "street" style [9],
- Automatic generator code creation with a schematic as the input that allows immediate generation of non-hierarchical matrix-style layouts [10].

Based on the previous advances mentioned above, the key contributions of this work are the following:

- The new method creates generator code from an input schematic and automatically links layout instances to a variety of templates in order to control them.
- Via a parameter mask, placement patterns, instance rotation, and routing channels can be defined by the user through adapting (template) parameters.
- The new method creates hierarchical generators, each of which incorporates the aforementioned templates to ease layout flexibility through hierarchical composition.

As a result, our presented method allows the translation of a hierarchical schematic into an executable and hierarchical generator, which immediately provides several place and route options among a set of pre-defined place and route templates.

This way, early layout extraction can be carried out in order to analyze layout behavior very early in the design flow and accelerate design. Further manual rework of the generated layouts is fully supported, as the approach creates persistent cells and views in the design library.

## 2. State of the Art

This chapter presents a brief overview of the state of the art in the automation of analog IC design. A distinction is made between optimization-based *top-down* and procedural generator-based *bottom-up* approaches [5] (We adopted the notations *top-down* and *bottom-up* from [5], as this summarizes the methods' natures in a concise form. However, it should be noted that these terms address the major nature while either of the methods might still utilize elements of the respective other approach.). We believe that templates are promising because they combine both methodologies (Figure 2). The respective advantages and disadvantages are discussed in the following subchapters.

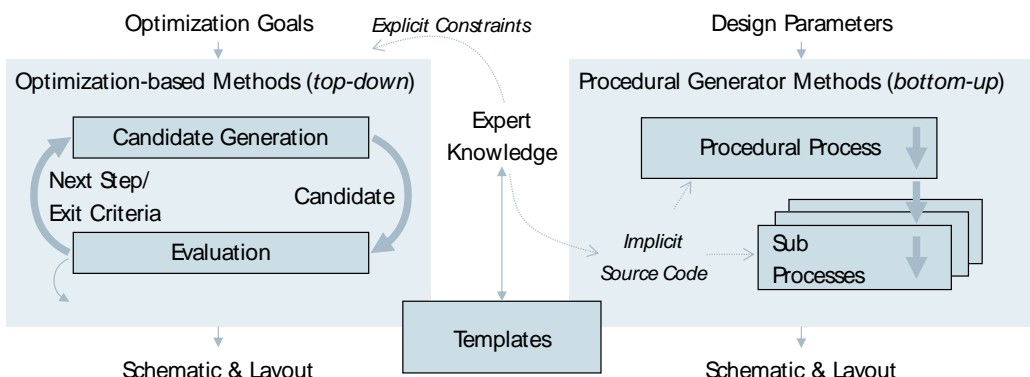

**Figure 2.** Comparison of top-down optimization (**left**) and bottom-up procedural generation (**right**). While optimization-based approaches search a solution that fulfills the given set of constraints and that improves the performance, procedural approaches execute a pre-defined design strategy in form of generator source code that is based on expert knowledge. This figure is adapted from both [5,6] to which we added templates as our proposal to link both methods with each other.

### 2.1. Optimization

Optimization-based approaches are already available in commercial design environments for schematic sizing. Major design software vendors such as Cadence® (San Jose, CA, USA), Synopsys® (Mountain View, CA, USA), and Siemens EDA (Munich, Germany, former Mentor Graphics®) offer tools that specifically "try" sizing variants of manually designed circuit topologies and, thus, automatically optimize them. In addition, companies such as MunEDA (Unterhaching, Germany) [11] and Intento Design (Paris, France) [12] have specialized in circuit optimization and analysis, respectively. In practice, optimization methods tend to be applied to smaller building blocks and components (e.g., standard cells or operational amplifiers), since larger circuits such as converters require too much

computation time when optimized. Approaches to more systematic and multi-disciplinary design approaches are, therefore, the subject of further research [5,13–15].

In the area of optimization-based layout automation, there is still mostly academic work. In some cases, these are part of larger frameworks and, thus, also consider parasitic layout effects during circuit sizing (often called "layout-aware sizing"). Such approaches use so-called templates (see next subsection) [16–19], explore templates during runtime [20,21], or even synthesize layouts without the help of templates [3]. In refs. [22,23], SWARM is presented which is an iterative method of self-organization for layout placement and wiring based on explicit and implicit specifications. With the recent rapid development of machine learning (ML), there are also ML-based optimization methods that can provide better results than previous optimization-based algorithms even at fewer iterations [24]. Pulsic Animate™ (Bristol, UK) [25] is a commercial tool for (partial) automatic layout synthesis based on a given hierarchical schematic. The typical circuit complexity is mostly in the spectrum of components, such as operational amplifiers or smaller building blocks.

*2.2. Templates*

Template-based layout methods are often used in the optimization-based approaches mentioned above. Templates restrict the solution space of the optimization problem. They are a special kind of constraint for specifying the floorplan and sometimes the routing of analog blocks in a knowledge-based way. The underlying method of template description, its implementation, and solving strategy (e.g., by evolutionary approaches) is the subject of research [26–30]. As an example, the framework AIDA [13] utilizes templates and combines many academic works, from optimization algorithms to layout automation used for various designs. Commercially, templates are used both by Jedat Inc. (Tokyo, Japan) for the automation of analog component layouts and basic building blocks [31] and by Synopsys® in their "Visually-Assisted Automation (VAA)" [32].

Across the literature, however, the template methods differ. Therefore, [33] first explicitly distinguished *symbolic* and *geometrical* templates. Symbolic templates are explicit geometric constraints, whereas geometric templates correspond to a parameterizable procedure (or "procedural generator"). According to the definition of our work, the geometric templates correspond to procedural generators. In ref. [16], for example, the term "template" is used for a complex PCell with a pre-defined (programmed) arrangement. There, the template is merely the (graphical) representation of the layout arrangement implicitly programmed by an expert into the procedural generator code. Thus, we would classify it as a procedural generator that implements a static template.

*2.3. Procedural Generators*

In contrast to optimization-based approaches, the expert knowledge contained in procedural generators is not available explicitly (i.e., it is not machine-readable), e.g., in the form of constraints. The expert knowledge is *implicitly* "hidden" in the procedural source code (Figure 2). Procedural generators are, therefore, interpretable and executable, but the (implicit) decision paths they contain—the expert knowledge—are executed directly without being interpreted by the machine. The procedural generator, thus, does not "understand" the intention of the source code. With this implicit way of implementing generators, it is hardly possible to identify the described structure other than by analyzing the generator source code with a "keen eye". If not using a dedicated API such as in refs. [6,7], there is no possibility to automatically extract abstract information of the layout arrangement from the procedural source code (e.g., as a return value of a method).

Furthermore, the (many) parameters used are very diverse and allow flexibility only according to the pre-programmed sequence, e.g., with respect to topology variants, sizing, or layout specifications. The complexity of the source code required for such flexibility quickly leads to high development efforts and costs, while such generators are difficult to maintain. For this reason, several methods were proposed that would diminish this shortcoming. On the schematic level, PCDS [34] reduces the number of code lines that lead

to the intended schematic creation results. For optimization, LAYGEN II improves device-level generation [26]. With the Berkeley Analog Generator (BAG) [4], a first open-source attempt covering a variety of target PDKs was presented. A second version of the BAG followed [35], and new layout engines such as MESH [8] for regular array-style layouts or LAYGO [36] to especially support gridded FinFET layout styles were proposed.

The underlying trend we observe is that the level of abstraction increases through additional layers of ever more high-level layout description. This way, details are encapsulated, and rather the "what" than the "how" is implemented, i.e., we see a transition from implicit procedural approaches toward more explicit declarative approaches. We believe that in the long run, this is the pathway to combine procedural generators with optimization towards comprehensive synthesis methods.

## 3. Materials and Methods

Some of the underlying materials and methods used are covered by NDAs with semiconductor manufacturers. This includes details such as layout design rules, detailed layout information, or device parameters. In addition, the generator tool presented is proprietary. However, it can be made available upon request, for example, together with the technology setup available for the GPDK45 from the Cadence® support website [37].

Following, we describe the materials and methods used. For further detail, Figure A1 in Appendix A shows an excerpt of the automatically created generator code.

### 3.1. Generator Approach

#### 3.1.1. Generator Framework

Details of the generator framework used are covered in ref. [7]. The underlying concept is that persistent DRC-clean and LVS-clean cells are automatically generated from a single source generator through parametric and procedural code. Thus, a generator creates at least schematic, symbol, and layout for any given (building) block.

Key extensions towards (matrix-style) templates have been presented in refs. [8,10], of which the latter also creates generator code automatically for non-hierarchical matrix-style blocks. These extensions have been further developed in the presented work and now support both hierarchical blocks and further template styles.

#### 3.1.2. Building Block Generators

The building block generators used are still implemented in an entirely procedural fashion. This means they do not yet rely on template-based generator code and are, thus, based on comprehensive source code developed by expert generator developers. Our library of building blocks especially contains blocks for (1) transistor arrangements, (2) capacitor arrangements, and (3) resistor arrangements. Examples of generated transistor building blocks for both a current mirror and a differential pair are shown in Figure 3. All generators can be parameterized not only regarding device sizing but also regarding both topology variants (e.g., differential pair vs. current mirror) and layout arrangement (e.g., number of rows, placement pattern, routing options, or dummy devices) in a relatively flexible way. As the generators cover all relevant views (schematic, symbol, and layout), schematic-level design entry using generators already provides parameterized building block layouts.

### 3.2. Generator-Embedded Templates

Our new approach embeds templates into generators, especially at higher hierarchy levels. In this paper, template stands for the *machine-readable* and *symbolic* representation of a layout, independent of the concrete (computational) representation. A template, thus, does not exactly replicate the actual layout but specifies the constraints for its design. So, templates are abstract specifications for layout and, thus, independent of a specific technology. Figure 4 shows an example of a template represented by both a floorplan and a slicing tree, which have already been used in "template"-based analog layout synthesis [16].

Slicing trees model slicing floorplans using binary trees. These trees are graphs that fan out from the "root" towards the "leaves", as often used in EDA [38].

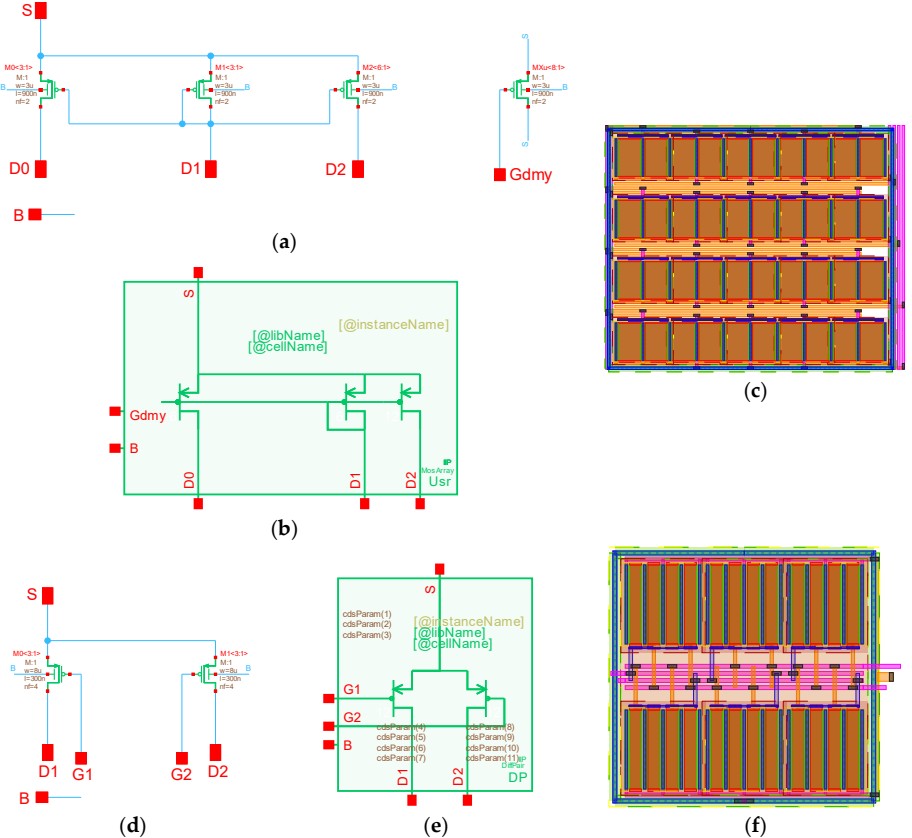

(a)

(b)

(c)

(d)

(e)

(f)

**Figure 3.** Examples of views of building blocks generated by procedural generators in a 22 nm technology (the views were converted to SVG files from original design data, thus, minor variations can appear). Above, the (**a**) schematic, (**b**) symbol, and (**c**) layout views of a current mirror with two branches and dummies are shown with a unit transistor sizing of two fingers, a width of 3 μm, and a length of 900 nm. Below, a differential pair is depicted in (**d**–**f**), respectively without dummies and unit devices with four fingers, a width of 8 μm, and a length of 300 nm, each.

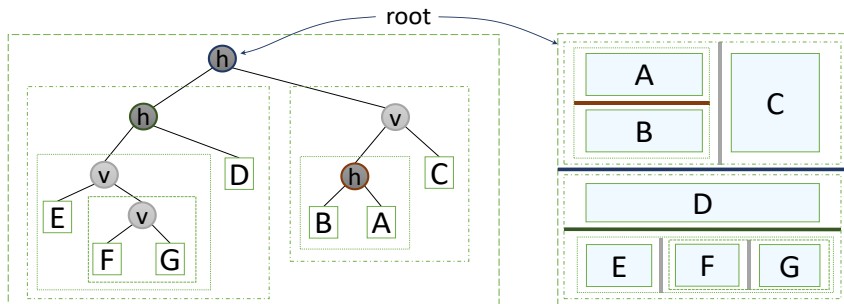

**Figure 4.** Representation variants of a template example. The right representation shows a floorplan, while the left one depicts the associated slicing tree. In addition, differently dashed lines indicate corresponding hierarchy levels and the colors of the horizontal cuts (*h*) show how both representations map to each other. The figure is adapted from [16]. © 2008 IEEE. Reprinted, with permission, from Castro-Lopez, R. et al. An Integrated Layout-Synthesis Approach for Analog ICs, IEEE Transactions on Computer-Aided Design of Integrated Circuits and Systems, 2008.

Specifically, we propose two major templates, which we believe can cover a majority of layout arrangements when utilized hierarchically. They represent layout arrangements

in a *matrix* style as well as in a *street* style. While the matrix style represents arrangements using mathematical matrices, the street style implements an upper instance row and a lower instance row connected via a central routing bus. The latter is represented by tuples of instance identifiers as well as net names.

Besides such regular templates, non-regular (more flexible) templates can be defined. They can be categorized into slicing floor plans and non-slicing floor plans [38–40]. However, either of them is not considered in the design methodology presented in this paper. The reason is that by using regular templates, we can (1) implement the templates straightforwardly using object-oriented source code without the need of a solver, as in ref. [30], and (2) we can easily include routing channels represented as placeable pseudo instances in an object-oriented way that allows flexible adaptation of their size through hierarchical composition using the composite design pattern. The template styles are depicted in Figure 5.

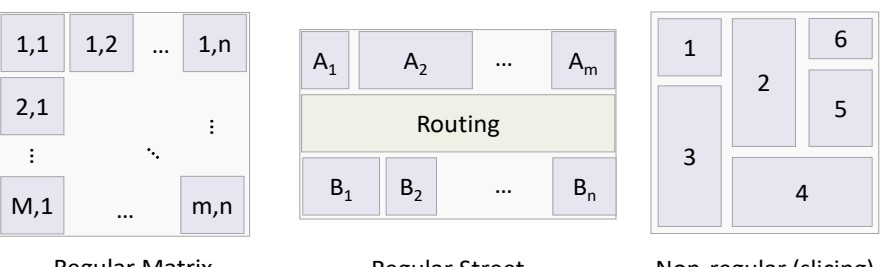

Regular Matrix　　　　　　　　　Regular Street　　　　　　　　Non-regular (slicing)

**Figure 5.** Comparison of exemplary template styles. The matrix and street templates are very specific and can, therefore, be addressed by means of indices. Non-regular templates, on the other hand, can specify almost any arrangement (here: slicing arrangement), but they require a method to resolve the abstract placement definition. Reprinted from [41], with permission from VDI Verlag, from Prautsch, B. Layout-Generatoren für den Analogentwurf in kleinen Technologieknoten; Fortschritt-Berichte VDI, Reihe 20, Nummer 478; Dissertation, TU Dresden, Dresden: Germany, 2022.

### *3.3. Generator Code Creation*

Designing generators is time-consuming and can, depending on the circuit size, easily require weeks to months of programming. Thus, we developed automatic creation of generator code with the schematic and symbol as the input. As output, a generator (i.e., its source code) is automatically created that parametrically (re-)creates the input views. In addition, it also creates a layout view. Thereby, the layout creation process is controlled by the template system which is embedded into the generator code.

#### 3.3.1. The Generator Programming Interface (API)

The generator programming interface implements generators in a common way that inherits from a generator parent class. Each generator follows the same structure and implements the following methods (see an excerpt of the generated code for a high-pass filter (HPF) example in Appendix A which contains the following methods and instance identifiers):

The procedure *param_check()* is used to define all parameters shown in the user interface. For this purpose, initial instances of objects might also be defined, including default settings for parameters. Each parameter has constraints attached, including choice constraints or range constraints.

The checking of parameters in the context of others happens in method *param_check()*. Here, callbacks are defined that run at any parameter change in order to, e.g., derive parameter values from each other, (de)activate parameter entry, or raise error messages to send hints to the user interface so that users can react.

In order to unify the parameters across the views, method *prepare()* collects view-overarching information such as instances, nets, or terminals (pins). This way, subsequent procedures can use the pre-defined information via methods which systematically reduces the likelihood of LVS issues.

The actual view definitions of the cell to be generated are provided in methods with the respective view name. Thus, schematic, symbol, and layout are implemented in individual methods. They might be deselected by the user in order to exclude them from the generator run if not required.

Please note that the integration of the templates is implemented via the object instance *self.tpl* (see Figure A1 in Appendix A) already defined in the method *param_check()*. This allows for user-driven interaction and immediate feedback (e.g., matching patterns or error messages). Once the parameters are defined and the user runs the generator, the template responses to the instance details that are available after instance generation. This way, the template adapts and executes placement and routing of the subblocks.

### 3.3.2. The Generator Creator Flow

The Generator Creator translates a given hierarchical schematic-level design into a hierarchical template-based generator which also generates the layout view. The flow with its major steps is shown in Figure 6. From a user's perspective, a separate tool is run, whose default settings are set in such a way that the behavior presented in this paper is immediately executed. Further options, for example, allow instances to be handled as they are (i.e., they are not converted into a sub generator).

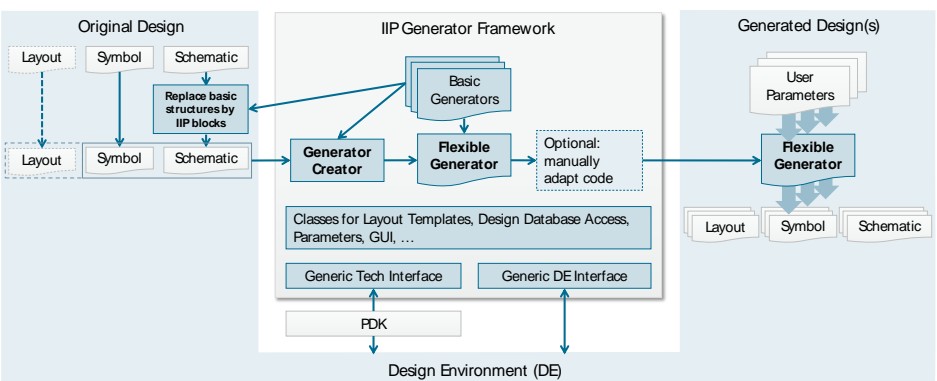

**Figure 6.** Flow chart of the Generator Creator flow. Given a design as the input, the Generator Creator analyzes and abstracts the given information through the Generator Framework. In this step, also the PDK-related information is mapped to a generic representation. With the generator code automatically created, the user can immediately run the generator. The generation procedure also includes the layout view which is parameterized through the template system.

### 3.4. Result Quantification Approach

A major challenge in comparing analog EDA approaches is the lack of benchmarks. To the best of our knowledge, there is no common quantification measure defined that allows for comparing our method against other methods. Therefore, we combine *quantitative* measures, including the number of code lines, generator run time, and the number of circuit instances with *qualitative* measures, such as the nature of the method, properties of the method, and usability aspects. Despite not being optimal, we believe that this is the best attempt to treat the lack of benchmarks (as well as the heavily NDA-restricted environment). Future work should elaborate further on the aforementioned limitations.

## 4. The Method and Results

### 4.1. The Example Circuit

In order to demonstrate the method, we selected a simple high-pass filter (HPF) as the input schematic. As the filter behavior depends on the parameters of the passive devices, this is likely a good example of a recurrent and parametric design task. Other relevant circuits could be blocks such as LDOs for various loads, different pipeline stages for data converters, or operational amplifiers.

The example circuit consists of a simple first-order RC high-pass followed by an amplifier with resistive feedback. It is implemented with the building block generator approach mentioned in Section 3.1, resulting in the schematic given in Figure 7 (with a block diagram given, too). The amplifier in the HPF is also implemented using generators. As all building block generators create the corresponding layouts (besides schematic and symbol), the respective building block layouts are already immediately available during the schematic design entry. At this stage, they are still unconnected when changing to the layout view (Figure 8). Nevertheless, the generators encapsulate device-level details such as matching placement patterns, routing layers, wire sizes, and substrate connections.

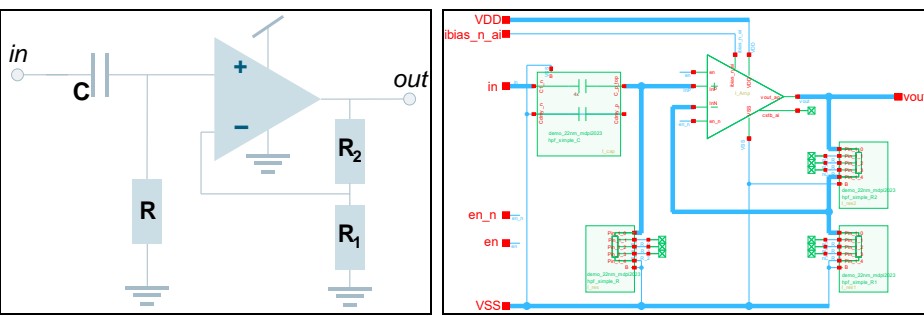

**Figure 7.** Topology of the simple high-pass filter (HPF) consisting of an amplifier, resistors, and a capacitor. The block diagram is depicted on the left and the corresponding schematic diagram (vector graphic which is derived from the actual design view), that includes generated building blocks for both amplifier and the passives, is shown on the right.

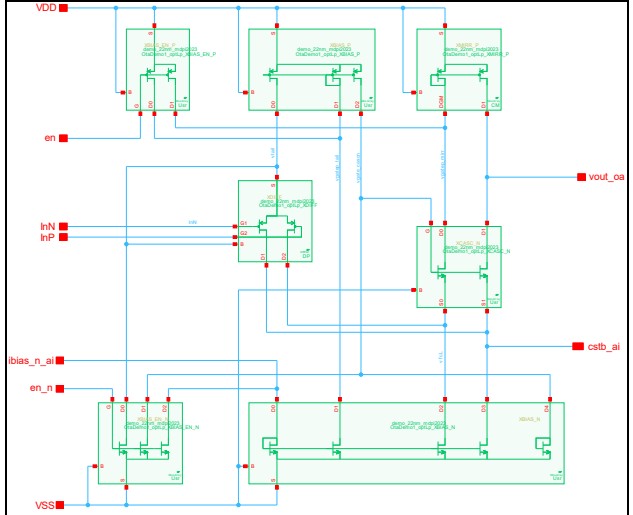 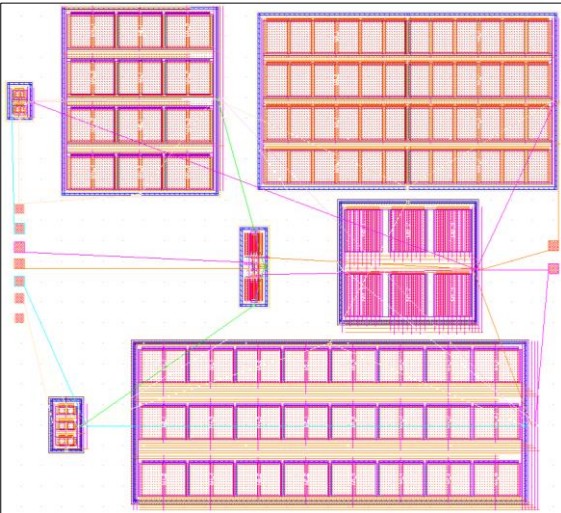

**Figure 8.** Amplifier with the building blocks generated. The schematic is depicted on the left (vector graphic which is derived from the actual design view) and the initial, unconnected layout with full building block layouts is depicted on the right (manually placed according to the schematic positions). All building block layouts are automated by generators based on layout parameters that can be controlled already during schematic-level design entry.

*4.2. The Generator Creation Process*

Given the generator-based hierarchical schematic with automated building blocks described in Section 4.1, the Generator Creator flow is executed. It creates a new top-level generator based on the schematic input. In addition, this generator embeds the template method described in Section 3.2. As a result, a new generator is available which not only parametrically (re-)generates the input schematic but also generates an LVS-clean layout according to user inputs regarding placement pattern (within the limits of the template

selected) and routing parameters (i.e., layers, widths, and spacings). This flow from schematic to new generator is fully automated and does not require any programming. The steps of the generator-creator flow are depicted in Figure 9 (which refines Figure 1). In order to reach a LVS-clean result, a few generator runs might be required in order to fine-tune the parameters of both the building block generators and the top-level placement. The steps of the whole process are as follows:

1.  First, the schematic entry is completed in a manual design fashion, starting with an empty schematic. Generators are used to encapsulate basic building blocks that also allow defining details of the building blocks mainly including device sizing. In addition, it also includes proper parameterization of the layout generation process such that the shapes of the building blocks, their rotation with respect to instance pins as well routing options are prepared for block assembly on the level above.

2.  Second, the actual code creation step is executed via a user interface that is accessible in the design environment (here: Cadence® (San Jose, CA, USA), Virtuoso® (San Jose, CA, USA)). The code creation step automatically runs the following steps:

    a.  Fetching both schematic and symbol information.
    b.  Mapping of technology-related information to a generic representation using technology abstraction [6].
    c.  Stepwise creation of the generator code sections according to the common generator structure given in Section 3.3.1. Here, the PDK-agnostic technology abstraction layer mentioned above is used, and the instances found are assigned to the template system (see Section 3.2) such that the templates are both accessible from the generator's user interface as well as used to control the layout generation routine.
    d.  Creation of the whole generator file and write-back into a user-accessible generator library.

3.  Thid, in order to create the layout, the automatically created generator can be run immediately to use it in a similar way as the building block generators before:

    a.  The new generator inherits the parameters of the sub-generators and, thus, provides full parametric control across the hierarchy.
    b.  Each hierarchy level incorporates templates for placement and routing options in order to flexibly define placement patterns, e.g., the order of wires in the routing channel.
    c.  The actual layout, including schematic and symbol, is (re)generated according to the user's inputs.

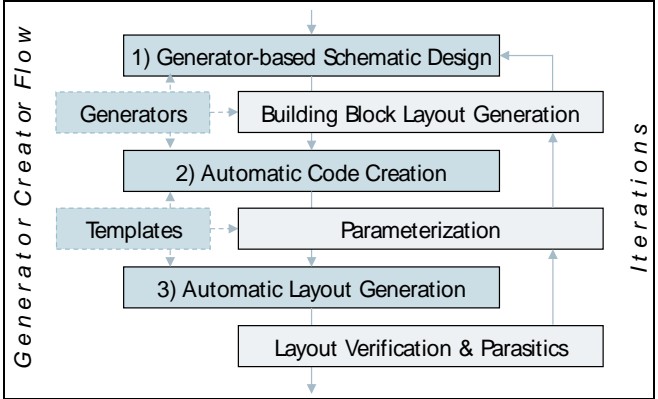

**Figure 9.** Depiction of the Generator Creator flow. The basic steps include generator-based schematic entry, automatic code creation, and automatic layout generation. During these steps, both already existing procedural building block generators and flexible templates are utilized. Based on the generated results, iterations might be required in order to adapt (sub) generator parameters.

### 4.3. User Interface and Generator Parameterization of the Placement Pattern

When running the automatically created generator, various parameters across the generator hierarchy can be defined. Besides device-level details such as sizing, placement, and routing, these can be controlled through the user interface. The basis for this is the flexibility of the respective template that was selected (here: street).

The relation between parameters, template, and user interface is illustrated in Figure 10. The hierarchies of both amplifier "I_Amp" and high-pass filter (HPF) (both indicated by colors) can be edited through the top-level GUI. For example, the lower hierarchy level of the amplifier can be adjusted regarding the placement pattern by providing another tuple of lists for both placement rows in the street template. Similarly, the top-level pattern can be edited while the amplifier generates a hierarchy level below. In addition, various details of all building blocks can be changed (see Section 3.1.2).

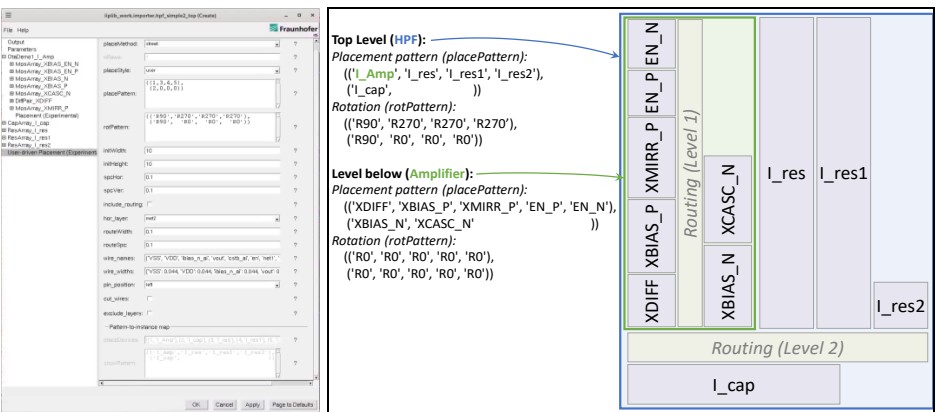

**Figure 10.** GUI of the hierarchical template-based generator (**left**) and the corresponding representation with illustrated hierarchy (**right**). Based on the hierarchical parameters for placement method (*placeMethod*, here the template called "street"), instance place pattern (*placePattern*), wire names (*wire_names*), and rotation pattern (*rotPattern*), the abstract template illustrated on the right is fully represented. Upon generation, the building blocks are generated and the predefined template controls the layout placement and routing process starting from the lower level generators including the amplifier towards the top-level generator.

### 4.4. Layout Variant Generation and Simulation

Using the automatically created generator, we generated different layout variants of the amplifier and a simple HPF that includes these amplifiers. The HPF placement is defined according to the abstract arrangement given in Section 4.3. All placement variants are intended to test both the layout-level flexibility and the hierarchy support of the method. Additionally, it is to identify the runtime as well as the limitations of both the building block generators (that do not yet utilize the presented template method but full procedural programming) and the automated generator.

The amplifier variants implemented are based on two different parameter sets: low power, "LP", and higher bandwidth, "Speed". Layout details of the amplifiers are shown in Appendix B. There, it can be seen that both the sizing and arrangement patterns of the building blocks as well as the overall arrangement can be controlled by the generator in a flexible way (trough adapting the template). While Figure A2a,c show the street arrangement with two rows (note that in these cases the arrangement is rotated by 90°, thus, the rows of the templates appear as columns), Figure A2b,d show each of them with one row.

Even though this paper does not focus on the particular circuit but on the EDA method, simulations were run in order to help evaluate the method. The generated amplifier layouts were simulated post-layout and compared with the schematic level. Table 1 lists these simulation results and shows the deviations of the layout variants related to the respective

schematic. However, it is not possible to derive generalized statements from it, as analog circuit performances largely depend on the context in which they are used.

**Table 1.** Comparison of schematic-level and post-layout simulations of the amplifier variants.

| Variant Measure | LP, Schematic | LP, Layout 1 | LP, Layout 2 | Speed, Schematic | Speed, Layout 1 | Speed, Layout 2 |
|---|---|---|---|---|---|---|
| DC Gain (dB) | 42.15 | 42.74 | 42.55 | 34.6 | 32.89 | 32.11 |
| *Deviation* | | *1.40%* | *0.95%* | | *−4.94%* | *−7.20%* |
| 3dB BW (MHz) | 0.770 | 0.781 | 0.777 | 2.20 | 2.19 | 2.30 |
| *Deviation* | | *1.36%* | *0.88%* | | *−0.45%* | *4.17%* |
| Phase Margin (°) | 78.56 | 75.71 | 74.45 | 83.4 | 83.17 | 83.45 |
| *Deviation* | | *−3.63%* | *−5.23%* | | *−0.28%* | *0.06%* |
| DC Current (μA) | 89.5 | 92.2 | 91.5 | 253 | 248 | 259 |
| *Deviation* | | *2.93%* | *2.16%* | | *−2.09%* | *2.29%* |
| Settling, rise (ns) | 162 | 160 | 152 | 110 | 111 | 112 |
| *Deviation* | | *−1.54%* | *−6.11%* | | *1.27%* | *2.27%* |
| Settling, fall (ns) | 123 | 144 | 152 | 106 | 110 | 110 |
| *Deviation* | | *16.88%* | *23.13%* | | *3.40%* | *3.96%* |
| Offset (mV) | 0.812 | 0.930 | 0.753 | 5.83 | 19.2 | 25.9 |
| *Deviation* | | *14.57%* | *−7.25%* | | *228.76%* | *344.29%* |

In order to further evaluate the hierarchical approach, both generated sizing variants of the amplifier were applied to the aforementioned HPF example from Section 4.1. Using the abstract placement pattern from Section 4.3, two layout variants were generated with the automatically created generator. The resulting layouts are shown in Figure 11. One can see both the different amplifiers at the upper left of the layouts as well as differences in the sizing of the capacitance arrays at the bottom, which is intended to get a more rectangular overall shape.

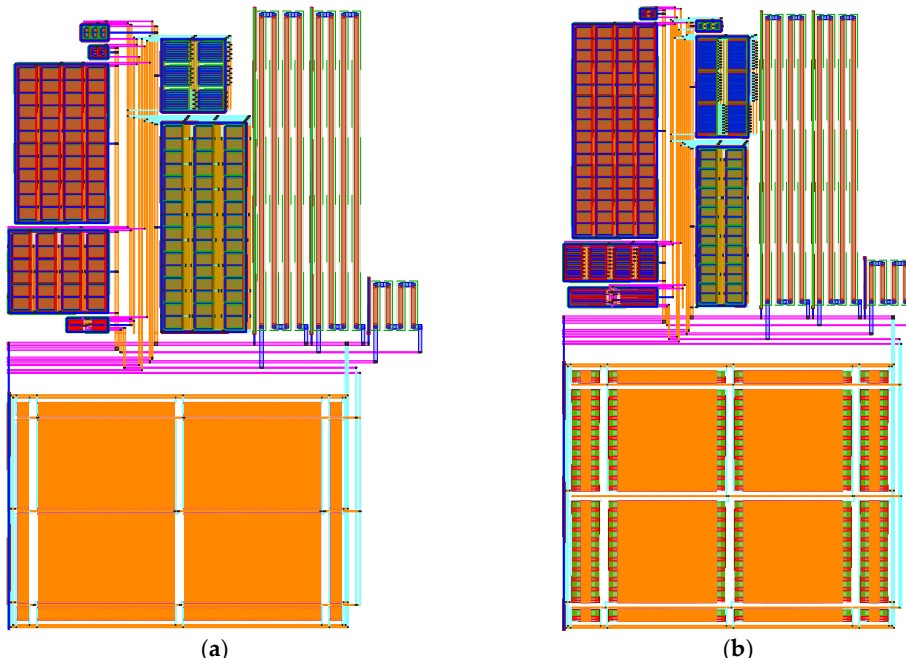

(**a**)       (**b**)

**Figure 11.** Generated layout examples of the HPF in a 22 nm technology based on the placement pattern given in Section 4.3 (vector graphic exports from the actual layouts, thus slight deviations might occur). While (**a**) includes the amplifier variant "LP", (**b**) instantiates the amplifier variant "Speed". In order to get a rectangular shape, the capacitor arrangement at the bottom was generated with adapted unit device sizing.

Both layouts were simulated. The frequency behavior is given in Figure 12, and further performances are listed and compared in Table 2.

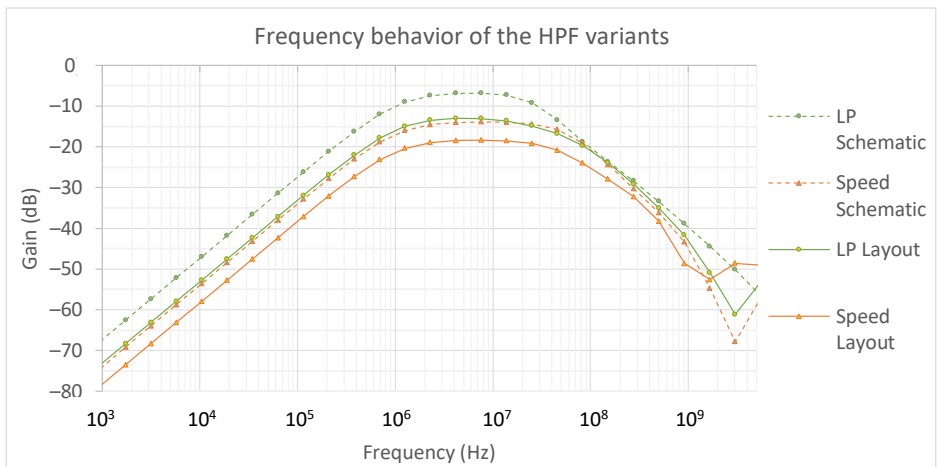

**Figure 12.** Frequency behavior of the HPF variants when using the amplifier variants "LP" and "Speed". Each variant was simulated both on schematic level and post-layout. It can be seen that the speed variant produces less gain but achieves a higher upper corner frequency. After extraction, about 5.7 dB and about 4.3 dB gain is reduced for variant "LP" and variant "Speed", respectively.

**Table 2.** Comparison of schematic-level and post-layout simulations of the HPF variants.

| Measure | Variant HPF with Amp LP, Schematic | HPF with Amp LP, Layout | HPF with Amp Speed, Schematic | HPF with Amp Speed, Layout |
|---|---|---|---|---|
| DC Current (µA) | 774.7 | 776.7 | 928.3 | 925 |
| *Deviation* | | *0.26%* | | *−0.36%* |
| Input capacitance (fF) | 989 | 1025 | 1060 | 1067 |
| *Deviation* | | *3.68%* | | *0.66%* |
| PSRR @ DC (dB) | 23.35 | 20.37 | 34.31 | 34.35 |
| *Deviation* | | *16.88%* | | *3.40%* |
| PSRR @ worst case (dB) | 23.09 | 20.36 | 28.28 | 26.74 |
| *Deviation* | | *14.57%* | | *228.76%* |

### 4.5. Quantitative and Qualitative Measures of the EDA Method

As mentioned in Section 3.4, comparing analog EDA methods is very limited. To the best of our knowledge, there are no benchmarks available for analog EDA. Thus, we evaluate the following aspects of the method: type of the generator, flexibility of the generator, lines of generator code, time to generator code, and cumulative generator runtime. The evaluation results are given in Table 3. All investigated runtimes were carried out on a server with a Xeon E5-2637v3 CPU.

While creating the test circuits, the parametrization of generators allows to both resize the building blocks and adapt their aspect ratios in a matter of seconds. On the level of the amplifier, this accounts for about 40 s, and the whole HPF takes about 120 s.

Running and adapting the generators works well when changing the placement pattern or sizing values. Limitations, however, occur if topology parameters are changed during the generation process (e.g., number of branches of a current mirror block). The reason for this is the change of the netlist, which is not yet flexible in the generator *after* automatic code creation. Therefore, the user cannot edit the netlist via parameters and, thus, either has to adapt the generator code or—more user-friendly—has to rerun the Generator Creator so that an updated generator is available with a new topology and netlist.

**Table 3.** Quantitative and qualitative comparison of the generators used.

| Generator | Type | Flexibility (Selection) | Approx. Code Lines | Time to Generator | Typical Runtime (Xeon E5-2637v3) |
|---|---|---|---|---|---|
| MosArray/ CapArray | Procedural; manually programmed | Transistor and Capacitor topologies; matching style & arrangements; layout details | 5500/2900 | Months (manually) | About 5–10 s (varies with arrays size) |
| ResArray | Procedural; manually programmed | Resistor series and parallel circuits; routing width | 1950 | Weeks (manually) | About 15 s |
| Amplifier | Template-based procedural; automatically created | Hierarchical block generation incl. sub generator parameters; P&R pattern | 4800 | **Seconds (this work: automatically)** | About 40 s |
| HPF | Template-based procedural; automatically created | Hierarchical block generation incl. sub generator parameters; P&R pattern | 5700 | **Seconds (this work: automatically)** | About 120 s |

*4.6. Comparison with other EDA Methods*

Comparing analog EDA methods is challenging, as, to the best of our knowledge, there are no benchmarks available yet (which would be a valuable contribution). Thus, we compare our method qualitatively with other existing analog EDA methods in order to get a picture of its features.

*Procedural approach:* Our method is a procedural generator-based approach comparable to the Berkeley Analog Generator (BAG) [4,35] when considering building block generators. Our method also allows the development of more complex generators [7], but the initial effort of generator development turned out to be critically large for large designs.

*Template approach:* Our method also incorporates the template approach in order to diminish the limitations of pure procedural approaches. Thus, the definition of the arrangement is comparable to the placement definition in AIDA [18] or the Layout Description Script LDS [28]. However, in contrast to AIDA and LDS, we do not utilize optimization to find a placement solution but rather heavily constrain the placement by the template. The template patterns "street" and the matrix style MESH [8] are implemented by the object-oriented composite design pattern such that all abstract positions can be directly addressed and instances attached. Depending on the instance sizes, the template elements are adapted, which aggregate along rows and columns from lower hierarchy levels to upper levels. Thus, all placements are inherently legalized, and routing is enabled by the pre-defined and also flexible routing channel. The hierarchical approach effectively forms a slicing tree.

*Optimization:* The presented method does not apply optimization. It is rather intended to be controlled by the user. Optimizers, such as WiCkeD™ from MunEDA (Unterhaching, Germany) [11], or sizing exploration, such as from Intento Design [12], could, however, be utilized by the user. Users can setup their individual optimization flows with scripts that parameterize the generator via the command line (which is another mode besides the GUI mode).

## 5. Discussion

*The problem class:* The presented work addressed the problem of the initial efforts required to set up analog integrated design automation. From the major analog EDA directions, namely synthesis and procedural generators, the latter was investigated in order to reduce the initial programming efforts.

*The solution concept:* The presented approach combines several existing methods into a novel user-driven flow. This differentiates our method from work that solely focusses on programming schemes or algorithms as support for EDA programmers. We attempt to solve the problem by using automatic code creation constrained by pre-defined templates.

*The scalability:* The templates can be selected flexibly via the user interface, which is automatically generated. In addition, they are implemented in a separate template system. This way, extensions in functionality do not require changes in the generator but only centralized updates in the template system. When utilized across a variety of generators, all (new) template functionalities (e.g., new template styles or algorithms for P&R) are automatically available across these generators. The code for typical layout design tasks is, therefore, well reusable, and maintenance is significantly eased by means of proper separation of data and procedure.

*The potential EDA community effect:* Through the separation of data and procedures, it is possible to provide the template interface definitions and related algorithms as open-source code. If adopted by the community, a (quasi) standard can evolve through a variety of individual contributions. This way, a library of templates can evolve, and benchmarks could be defined in a machine-readable way across EDA tools.

*The potential in analog IC design:* A major limitation of former analog layout generators is their initial development time. This often leads to the economically driven decision not to adapt them for productive design projects. To the best of our knowledge, our presented approach is the first to propose a fully user-driven way of creating hierarchical and flexible generators to overcome the need for programming generator code (which is sometimes also realized in a graphical way such as in ref. [42]). This sets the entry barrier for utilizing procedural generators significantly lower: in an iterative process, design engineers can design schematics using building block generators, parameterize them, and then run the Generator Creator to get a hierarchical generator immediately. By parameterizing the new generator's layout pattern, rapid layout prototyping is enabled, and early parasitic extraction will help design engineers taking decisions fast.

*The limitation to analog IC design:* So far, the approach cannot provide every possible placement or routing pattern, as it is limited to the capabilities the templates provide. Thus, specific requests of designers might not be covered yet, and manual edits based on the generated results will still be necessary. In order to improve the quality of the generated layout, the library of templates must be extended, ideally using a community effect. We strongly suggest to establish an analog EDA community for sharing automation attempts and actual code in order to join efforts on common ground that allows broader re-usability and adoption.

*The next steps:* Future research should refine the approach regarding the following aspects:

- The proposed method works best with schematics that, ideally, include generated building blocks. This, however, is in contrast with the "flat" schematic level design approach. Thus, user-driven conversion of flat designs into ones with generated building blocks instead should be considered.
- A source design might already contain a layout (e.g., when whole IPs are migrated). Therefore, algorithms should be investigated that translate static input layouts into abstract template representations. These templates would then guide the hierarchical layout generator automatically.
- The given set of templates is still limited and should be extended to include further styles. In order to ease the creation of a template library, an open approach should be provided, e.g., by providing an open-accessible API description.
- The presented routing scheme follows a straight-forward row-based pattern with a list of net names as the input. Algorithms that allow automatic and analog-aware routing are, thus, desirable and, ideally, open-access.

## 6. Conclusions and Outlook

This work presents a novel approach to combining procedural generators, flexible templates, and automatic generator code creation in a hierarchical way. As a prerequisite, a template-based approach for the explicit and flexible top-down description of generators was developed. Currently, two basic template styles are available, and further ones will be the topic of future work. The template approach increases the flexibility of otherwise relatively structure-static procedural generators, as respective strengths are supplemented and the weaknesses compensated. While procedural generators create layout details and also realize entire analog basic blocks automatically, templates allow flexible and abstract layout description and processing. The new combination of these methods allows both user-driven description and automation of hierarchical layouts in a fashion that does not require programming any code. With this method, the otherwise time-consuming programming of generators can be automated entirely, enabling rapid parasitic extraction early in the design process.

Our approach does not yet claim to produce best-in-class layouts. The focus, however, is on rapid generator code creation to provide designers early layout insights by means of almost instant layout automation. With this, clean layouts and parasitic extraction can be achieved in a matter of about 10 minutes to a few hours (when generators are used for schematic entry and both topology and initial sizing are available). In addition, the post-layout simulations show reasonable performance for the selected examples. Thus, we believe that our approach will enable designers both to improve decision-making in the early (schematic-level) design phase and to automate layout design steps.

As the templates are organized independently of the concrete generators, updates in the template system can be deployed to existing generators. This means that new algorithms at the level of templates will gradually improve the generated results (e.g., other placement and routing schemes, algorithms for parameter optimization, or estimation methods; in our example above, the routing scheme was updated this way). So, abstract and PDK-agnostic templates should become available to the research community in order to better spread knowledge in this comparably small and also highly NDA-restricted community and pave the way for more closely linked EDA research and developments.

**Author Contributions:** Conceptualization, B.P.; methodology, B.P., U.E., U.H.; software, U.E., B.P.; validation, U.E., B.P.; formal analysis, B.P.; investigation, B.P., U.E.; data curation, U.E., B.P.; writing—original draft preparation, B.P.; writing—review and editing, U.H., U.E., B.P.; visualization, B.P., U.E.; supervision, B.P.; project administration, B.P.; funding acquisition, B.P. All authors have read and agreed to the published version of the manuscript.

**Funding:** This research was funded by the German Federal Ministry of Education and Research (BMBF) within the frame of the HoLoDEC project, grant number 16ME0700. The article processing charge (APC) was funded by the Fraunhofer-Gesellschaft.

**Data Availability Statement:** Not applicable. In general, IC design is both very NDA-restricted and based on special closed-source design tools. Current activities towards open access IC design and EDA are, however, evolving and will be considered in future.

**Acknowledgments:** We would like to thank all our colleagues that contributed to the development of the former Generator Creator version which we extended such that it now supports templates.

**Conflicts of Interest:** The authors declare no conflict of interest. The funders had no role in the design of the study; in the collection, analyses, or interpretation of data; in the writing of the manuscript, or in the decision to publish the results.

# Appendix A

*Simplified Selection of the Automatically Created Code of the HPF Generator*

```python
# this is iiplib.imported.hpf
import iip…    # API
import iiplib…    # sub-generators
class Generator(iip.gen.HierBlock):
    # define parameters, their constraints, and init dependent class members
    # executed once when creating this generator object
    def param_spec(self):
        …
        # add sub-generators with default instance name and initial parameter values
        self.generators.add("I_cap", iiplib.base.CapArray, Params(w="1u", l="1u", …))
        self.generators.add("I_Amp", iiplib.std.OtaDemo1, Params(placeMethod="STREET", nRows=2, …))
        …
        # add constrained parameters
        #              param name      default value     doc string              optional constraints
        self.params.add("placeMethod",  "side-by-side",  "layout template type",    ChoiceConstraint((
            "side-by-side", "source", "MESH", "STREET")))
        self.params.add("nRows",          2,             "number of template rows", RangeConstraint(1, 2))
        self.params.add("placePattern", ((1,2,3,4,5,6),), "pattern of instance ids within template")
        self.params.add("cut_wires",    False,           "cut the routing channel wires at the last branch")
        …
        # add proxy parameters from sub-generators (hierarchical parameter propagation)
        self.params.add_proxy("I_Amp_nRows", self.generators.I_Amp.params.nRows)
        …
    # handle parameter dependencies, executed per parameter change
    def param_check(self):
        # configure layout template
        if self.params.placeMethod.v == "STREET":
            self.tpl = iip.placeroute.PlaceTemplateStreet(ncols=(3,3), route_opt=…)
            # assign existing instances, generator objects or placeholders to the template cells
            # doing this already here enables early area and aspect ratio estimation
            # even before generating layout data
            self.tpl.assign_elem(pos=(0,0), elem=self.generators.I_Amp)
            self.tpl.assign_elem(pos=(1,0), elem=PlaceTemplateElem("I_cap", width=5.0, height=5.0))
        …
    # common data for all views, executed once per generator run
    def prepare(self):
        # e.g. describe circuit structure/topology
        # correspondence of schematic and layout instance names
        #                     id      generator or master   sch inst name  lay inst name  iterated inst spec
        self.instnamespecs.add("I_cap", self.generators.I_cap, sch="I_cap",  lay="I_cap",  bus=None)
        …
        # net definition and port binding (netlist)
        #                  netname, (instname, instpinname), …             (termname), signal type,  bus spec
        self.netspecs.add("VDD",    ("I_cap", "VDD"), ("I_Amp", "VDD"), …, ("VDD",),   type="power", bus=None)
        …
        # terminal definitions
        self.termspecs.add("VDD", TermType.In, bus=None)
        …
    # schematic view description, executed once per generator run
    def schematic(self, cv):   # cv is the target schematic cellview
        # create instances
        i_cap = self.instnamespecs.I_cap.master.instantiate(cv, pos=Dot(0,0), rot=RotationType.R0, …)
        …
        # create wiring, pins, labels
        pin_vdd = cv.create_pin(self.termspecs.VDD, …)
        cv.create_wire(points=(pin_vdd, i_cap.find_pin("VDD")), routingType=RoutingType.ho_ve,
                       net=self.netspecs.VDD, …)   # rather explicit description, more generic are available
        …
    # layout view description, executed once per generator run
    def layout(self, cv):   # cv is the target layout cellview
        # create instances
        i_cap = self.instnamespecs.I_cap.master.instantiate(cv, …)  # instance of a generated block
        master = self.open_cellview("mylib", "mycell", "layout")
        i_2 = cv.create_instance(master, "I2", parameters=[…])  # instance of an existing (p)cell
        …
        # update the template with the real layout instances
        self.tpl.assign_elem(pos=(0,0), elem=i_cap)
        self.tpl.assign_elem(pos=(0,1), elem=i_2)
        …
        # draw to layout view
        self.tpl.draw(cv, …)
        …
```

**Figure A1.** Representative and simplified selection of the generator code that was automatically created by the method presented in this work. Note that the template *self.tpl* connects parameter entries of the user (method *param_check()*) with layout generation (method *layout()*).

## Appendix B

*Example Layouts Generated by the Amplifier Generator*

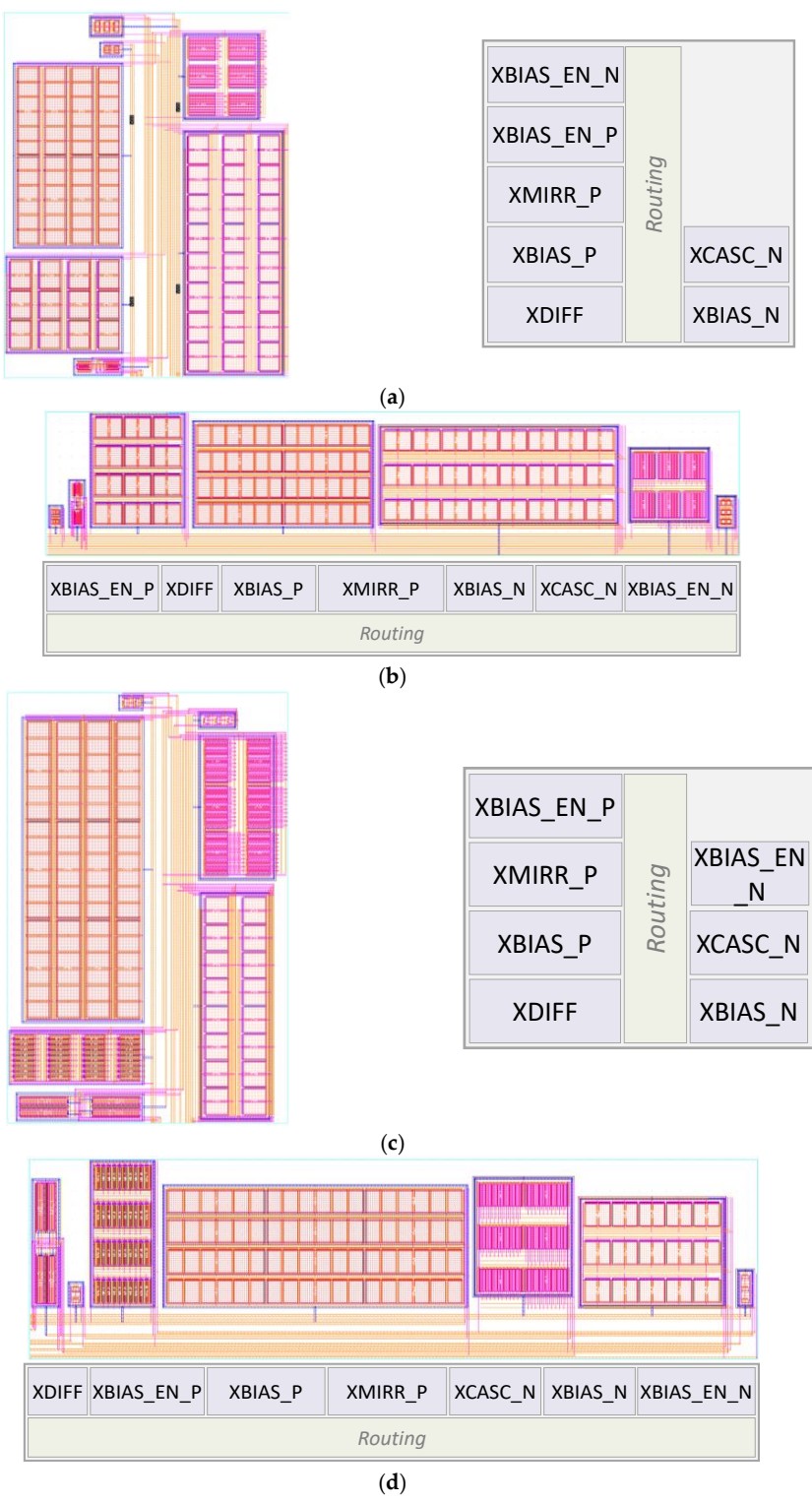

**Figure A2.** Generated layout examples (in a 22 nm technology) of amplifier variants by utilizing the presented method of template-based automatic generator code creation. Results for sizing "LP" are shown in (**a**,**b**) and results for sizing "Speed" are shown in (**c**,**d**). Besides sizing, the generated layouts are varied regarding placement patterns (street with one row and once with simpler routing vs. two rows and different arrangements each) and rotation patterns.

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
