# Peer review of "Generating the Generator: A User-Driven and Template-Based Approach towards Analog Layout Automation"

_electronics, doi:10.3390/electronics12041047_

Round 1

Reviewer 1 Report

This paper presents an analog layout automation approach based on templates targetd for usability.
The work seems promissing because it is based on a template approach, relieffing the user for a time-consuming code programing.
The generation approach links building blocks with the layout, using a template library, which is benefical for the usability and for end-users.
Moreover, it makes a review of approaches and works.
There are few remarks that must be addressed to allow the reader to compare this approach on their several solutions, and, why not (?), with solutions obtained from other approaches.
The layouts in Figure 3 are very raw and must be better described, e.g., which are the (W/L) rations for individual MOSFETs and the combined (W/L).
The layuts in Figure 11 are regulat matrix, regular street or slicing? It is not evident.
Rewrite Figure 7, more specifically the figure on right, it is hard to see what is what. Moreover, make this schematic from the scratch using a drwaing tool like Visio. Avoid placing schematics from the EDA tool, or at least complement with a good schematic.
The same applies to Figure 8. Give more details about the ceXMIRR_P cell.
With respect to the layouts of Figure 11, it is important to make post-simulations for each layout and compare with the ideal simulation from the schematic. Make a table with the most significant results and checking if the specifications are met.
Moreover, make a table, detailling all ellements (finger numbers + (W/L) of MOSFETs, number of elements of resistors, resistances, and so on) to allow the reader to make a full comparision of the solutions.
Line 402-403 are strange: ".... from a) to c) and c)...", what you mean with this?
To finish, it was interesting to include a table in the conclusions sumarizing the main features of this appproach with the listed from the state-of-the-art (AIDA, etc.) and commercials (Munich EDA, etc.).

Author Response

Dear reviewer,

We are currently working on the requested improvements. Specifically, we did the following (but not yet uploaded, due to work in progress):

Figure 3: The figure was changed with schematic and symbol added each plus ng, w, and l given in the caption.

Figure 7: The block diagram on the left already reflects the schematic on the right. By using the generators, it might be harder to read the schematic but this potential drawback should not be hidden behind another drawing. The caption was changes to make this more clear. Also, the schematic was adapted to better see the relation between block diagram and schematic.
Figure 8: to make the topic more clear, we changed the specific building block layout to a view of all the unconnected sub layouts incl. connectivity flight lines that immediately root from the schematic entry. This – combined with an extended explanation of the building block generators in sec. 3.1.2 – should ease the understanding that initially (prior to generating the generator), all blocks are generated but unconnected at this step. The connection through the automatically generated generator with template happens afterwards.

Also, we are currently running extracted simulations to improve the quantitative evaluation of the Figure 11 layouts generated with our proposed method.

A feature-wise comparison with other methods is a good idea. We will add another sub section to complement the quantitative measures with qualitative measures (features), too.

Kind regards

Benjamin Prautsch

Reviewer 2 Report

This paper proposes a user-driven controller” of the state-of-the-art analog layout generators. Various type of templates are taken into account, including building blocks, matrix style, “street” style. The overall framework is concise and reasonable. However, there are still some concerns that need to be addressed before publication.

1. The user-driven method for generators is more like a engineering problems.

The proposed method creates generator code from the input schematic and the code drives the basic layout generators which already exists. Although this method further allows hierarchical generator creation, the innovation is still average.

2. In experiment, authors only show some basic information of the benchmark, such as runtime ,generation type ,and flexibility. However, all of these metrics are unconvincing in the quality of final analog layout. In analog layout automation, the quality of layout must be evaluated with post-layout simulation results.

3. Is this framework easy to be compatible with other layout generators?

4. Is the quality of layout determined by the performance of basic generators?

5. There are several format errors:

In page 6, the layout images in figure 3 have different resolution obviously.

In page 9, the schematic image in figure 7&8 is difficult to see, please adjust the image resolution.

In page 12, table 1, the bottom left corner of the table is missing part of the horizontal line.

In page 16, wrong position of double quotation marks in some titles in the references.

6. Brief preliminary of circuit sizing is unnecessary in section 2.1; the references [12-14] have low relevance to analog layout automation topic.

Author Response

Dear Reviewer,

Thank you for your detailed feedback. Here come the replies to your concerns:

1) Yes, it probably largely is an engineering problem. The code generation, however, not only combines pre-defined generators, it embedds our specific (non-pcell) generators into another newly generated generator that not only works hierarchically but also includes additional generator code for, e.g. routing and pin creation. But in the essence, it saves programming time.

2) We are currently running simulations and add post-layout simulations to better evaluate the layout quality. However, we also expect the user to be part of this “loop”, as we believe that only a mixture of both automation and user interaction will be a good way to go (as even digital ASCI design is not really “fully automated” but user-guided). In addition, common benchmarks do not exist to the best of our knowledge (would be valuable future work).

3) What do you mean exactly? The framework does utilize the PDK’s device pCells and could also include more complex custom pcells. However, we did not yet try a combination with, e.g., the BAG generators (which would be another engineering problem).

4) Yes, the layout quality is usually strongly determined by the performance of the basic generators. As a rule of thumb, this is more true the smaller the minimal technology dimension is – but here also the (local) interconnects come more into play. However, the next upper hierarchy also strongly affects the performance, as there might (or might not) be (very) sensitive parts. Thus, generalized statements are not possible and the user/designer should always be in the loop.

5)

Page 6 & Page 9: did you read the .pdf version? This seems to be a conversion issue. If you check the Word version, the resolutions should be the same (only the zoom level may differ).

Page 12 & Page 16: Thank you, this was corrected in the version that is currently being edited by us.

6) The sentence was shortened and references removed. More emphasis was drawn on multi-disciplinary approaches as further research. This is important, as the templates could pave the way to better multidisciplinary automation in analog layout. Thus, we kept ref. 14 inside as their “everything at once” supports the statement that more multi-disciplinary methods arise (even though they apply such method for pre-layout sizing – it’s about the methodology).

The version that we are about to upload tomorrow will include the points mentioned above plus pre and post extracted simulations in a table.

Kind regards,

Benjamin Prautsch

Reviewer 3 Report

This work proposes a user-driven and templated-based method to automate analog layout. The pros and cons of existing methods, optimization-based, template-based, and procedural generators are detailed.  The proposed method reduces the timing to generator dramatically. The template-based method increases usability, however, also requires more human intervention thus less automatic than previous methods. Through this trade-off somehow promises certain improvements over current methods, therefore, within the scope of this publication.

1. For all the layouts, please provide the detailed schematic and mark the corresponding transistors, critical signal path, etc. in the layout maps so the readers can rate the generated layout by themself.

2. Mismatch, parasitic capacitances, and resistances are critical to analog blocks, specifically affecting the opamp’s poles and zeros, therefore, the post-layout simulation is recommended to have a comparison. This is necessary to demonstrate the claimed usability.

3. critical figures like Fig. 7 and figure 8 need improvement. Blurred figures cannot convey any valuable information to the readers. 

Author Response

Dear Reviewer,

Thank you for your detailed feedback.

Right, our proposed method us user-driven, thus, opposing to other fully-automated methods. We believe that this will be most suitable for designers, as their concern with fully automated methods is always an issue that should not be neglected. Still, the templates contain algorithms that should improve over time so that the designer can, e.g., select among different placement and/or routing methods.

Here come the replies to your numbered concerns:

1) We have added schematic (and symbol) views to Fig. 3 including information on sizing. In Fig. 8 we changed the layout view to a one with all building blocks arranged similar as in the schematic plus colored connectivity flight lines.
2) We are currently running extracted simulations and will add the results in a comparing table (schematic vs. one or two layout versions for both opamp and filter). For mismatch, we will simulate offset and possibly differences in input capacitances that would be relevant for, e.g., SC circuits.

3) Did you use the .pdf? It seems that the automatic conversion blurred the .docx significantly. We will make sure to upload our self-created .pdf file.

The version that we plan to upload tomorrow will include the points mentioned above plus pre and post extracted simulations in a table.

Kind regards,

Benjamin Prautsch

Round 2

Reviewer 2 Report

Thank the authors for addressing the comments. The reviewer has no more suggestions. 

Author Response

Thank you very much for your reviews and your valuable hints to our paper!

Reviewer 3 Report

1) the authors are suggested to use Visio or other software which can generate vector figures to draw the schematics in Fig.3, 8 etc.

Author Response

Dear reviewer,

we have built a script that translates schematics and layouts to SVG files and used those to replace the former pixel graphic figures.

Kind regards,

Benjamin Prautsch